# Effect of Short-Duration High-Intensity Upper-Body Pre-Load Component on Performance among High-Level Cyclists

**DOI:** 10.3390/sports10030032

**Published:** 2022-02-27

**Authors:** Dmitri Valiulin, Priit Purge, Jarek Mäestu, Jaak Jürimäe, Peter Hofmann

**Affiliations:** 1Institute of Sport Sciences and Physiotherapy, Faculty of Medicine, University of Tartu, 51008 Tartu, Estonia; priit.purge@ut.ee (P.P.); jarek.maestu@ut.ee (J.M.); jaak.jurimae@ut.ee (J.J.); 2Training & Training Therapy Research Group, Institute of Human Movement Science, Sport & Health, Exercise Physiology, University of Graz, 8010 Graz, Austria; peter.hofmann@uni-graz.at

**Keywords:** athletic performance, glycolysis, lactic acid, spirometry

## Abstract

The aim of the present study was to evaluate the effects of upper-body high-intensity exercise priming on subsequent leg exercise performance. Specifically, to compare maximal 4000 m cycling performance with upper-body pre-load (MPT_high_) and common warm-up (MPT_low_). In this case, 15 high-level cyclists (23.3 ± 3.6 years; 181 ± 7 cm; 76.2 ± 10.0 kg; *V**˙*O_2max_: 65.4 ± 6.7 mL·kg^−1^·min^−1^) participated in the study attending three laboratory sessions, completing an incremental test and both experimental protocols. In MPT_high_, warm-up was added by a 25 s high-intensity all-out arm crank effort to the traditional 20-min aerobic warm-up. Both 4000 m maximal bouts started with a 12 s all-out start. Heart rate, blood lactate concentration [La) and spirometric data were measured and analyzed. Overall MPT_high_ time was slower by 5.3 ± 1.2 s (*p* < 0.05). [La] at the start was 5.5 ± 1.5 mmol·L^−1^ higher for MPT_high_ (*p* < 0.001) reducing anaerobic energy contribution which was higher in MPT_low_ during the first and third 1000 m split (*p* < 0.05). Similarly, MPT_low_ maintained higher total average power during the entire performance (*p* < 0.05, *d* = 0.7). Although the MPT_high_ condition performed less effectively due to decreased anaerobic capacity, pre-load effect may have the potential to enhance performance at longer distances.

## 1. Introduction

Metabolic conditioning is an emerging trend intended to improve basic warm-up effects. Competition puts the greatest strain on all the body’s functioning mechanisms, while requiring the biggest effort from both local and systemic levels. ATP turnover efficiency rises with tissue oxygenation rates, and higher oxygen (O_2_) concentration allows the possibility to use slow but high-energy density substrates such as fat or intracellular lactate to provide ATP [1]. Exploiting metabolic mechanisms such as priming can provide higher O_2_ levels, speed up *V˙*O_2_ kinetics, and increase power output and submaximal activation of aerobic mechanisms [2,3]. Although higher warm-up intensities have been shown to produce larger priming effects, the selected intensity should not induce fatigue. Therefore, to optimize performance depends on the warm-up intensity, individual factors and recovery time between the warm-up activity and maximal performance [4,5].

In 1972, Klausen and colleagues [6] investigated whether earlier activation of aerobic mechanisms could provide a benefit to significantly change cycling performance. Decades later, successful speeding up of *V˙*O_2_ kinetics has been investigated in numerous studies; however, only marginal performance-related changes have been detected [7,8,9,10]. Practical implications should be therefore based instead on standardised approach; while improved physiological changes do not guarantee improved competitiveness [11].

A traditional warm-up process, consist of light endurance activity and stretching, is the basic requirement for successful performance [12,13,14]. In addition, a high-intensity sport-specific warm-up can improve 5-min cycling performance [15]. Commonly used warm-up techniques are based on isolated preparation of performance related muscles added to a general warm-up. In contrast, more general warm-up approaches can impact muscles globally without raising local muscle acidosis or using glycogen depot [16]. Accordingly, some potential is seen in a metabolic pre-conditioning approach that could be achieved by non-sport-specific muscle high-intensity pre-load [9,16]. Such priming is suggested to improve performance efficiency while keeping sport-specific muscles unaffected and in readiness. Müller et al. [9] have highlighted the opportunity to enhance target muscles’ aerobic work capacity by partial inhibition of lactate production during the first minute of exercise, which is suggested to speed up *V˙*O_2_ kinetics in a subsequent main workout. Previously, a variety of studies have investigated such priming effects by non-dominant muscles, e.g., leg exercise priming for subsequent arm and upper-body exercise [9,16,17,18]. However, physiologically beneficial performance enhancement could only be detected in small-muscle-group exercise [17]. Furthermore, Purge et al. [16] claimed a specific priming time schedule, the pacing strategy, and the fact that in the case of rowing priming cannot be performed in non-dominant muscles as the sport is a whole-body exercise. Interestingly, the specific physiological effect of lactate increase inhibition was not supported in that study.

In addition to the above mentioned physiological priming effects the pacing strategy may also be an important factor which needs to be addressed. It was shown that pacing can offer benefits when performing high-intensity exercises during the start phase in order to ensure adequate resource distribution throughout the distance [19,20]. Brock et al. [21] successfully combined all-out start pacing and previous priming, both executed by lower-body effort. Bohnert et al. [2] investigated upper-body pre-loading, which is qualitatively comparable, although lower in magnitude. The aim of the current study was to compare the effect of a usual warm-up compared to a high-intensity arm-crank priming exercise protocol added to the warm-up on 4000 m cycling performance. To guarantee equal perception of readiness a protocol according to Brock et al. [21] was applied.

## 2. Materials and Methods

### 2.1. Subjects

Fifteen well-trained male cyclists (mean ± SD 23.3 ± 3.6 years; 181 ± 7 cm; 76.2 ± 10.0 kg; *V˙*O_2max_: 65.4 ± 6.7 mL·kg^−1^·min^−1^) (Table 1) volunteered to participate in this study, which was approved by the local university ethics committee (290/T-17) in accordance with the Declaration of Helsinki. All subjects were required to give their written informed consent at the first laboratory visit. Subjects were instructed to arrive at the laboratory in a rested and fully hydrated state at least 2 h postprandial. They were asked to avoid ingesting alcohol and caffeine 24 h before each laboratory visit. In addition, subjects were asked to refrain from maximal exercise for at least 3 days before each laboratory visit.

### 2.2. Experimental Overview

Subjects completed three laboratory sessions within a 3-week period, with each visit being separated by at least 3 days. On the first visit, subjects’ body composition was measured using DEXA, they were introduced to the testing equipment, and recorded individualized ergometer parameters for the subsequent testing trials. To exclude any health risks associated with the maximum stress, all subjects had to perform a specialist-supervised incremental cycle ergometer test. On two subsequent occasions maximal performance tests (MPTs) were performed. Gas exchange variables, heart rate and blood lactate (La) concentration were measured in all tests.

### 2.3. Incremental Test

On the first visit subjects completed a maximal incremental cycle ergometer exercise test starting at 60 W and with increments of 20 W every minute. The workload was increased until subjects were no longer able to sustain the load. Two gas exchange thresholds (VT_1_, VT_2_) were determined by an experienced researcher considering a disproportionate increase in *V˙*CO_2_ relative to *V˙*O_2_, ventilation breakpoints (V-slope), visual inspection of individual plots and respiratory-exchange-ratio (RER) value. The highest mean measured O_2_ during 30 s period was considered *V˙*O_2max_.

### 2.4. Experimental Test

In order to objectively compare performance in two maximal 4000 m cycling tests with common warm-up conditions (MPT_low_; Figure 1) or upper-body pre-load (MPT_high_; Figure 2), subjects performed both protocols in randomized order. Subjects were instructed to complete both maximal tests as fast as possible. MPT_low_ and MPT_high_ protocols started with a 20-min warm-up at 40% of *V˙*O_2max_. A 25 s high-intensity all-out arm crank effort (braking weight: 35 g·kg^−1^ body weight) on an upper-body hand-crank ergometer was added in the MPT_high_ condition. The 25 s time period was suggested to induce a lactate increase of about 8–10 mmol/L to induce a sufficient subsequent inhibition without negative side-effects of the induced acidosis [16]. In both testing conditions, subjects were allowed to recover from the warm-up until they reported readiness for the subsequent maximal performance test but still maintained elevated blood lactate concentration. Maximal performance bouts started with a 12 s all-out start. In this case, 10 s before beginning each MPT, subjects were instructed to take a standing position and adjust the crank angle to their preferred position, as it was documented during the first laboratory visit. Subjects were provided with a 3 s countdown before the 12 s all-out start. Following the initial 12 s all-out upright cycling, subjects were instructed to take a seated position for the remaining test duration. Strong verbal encouragement was provided during both maximal tests. Remaining distance was the only landmark that cyclists were provided, and subjects were unaware of the elapsed time or implemented power (W), the power display and time were covered to exclude any numerical comparison.

Blood [La] concentration was determined for both conditions at rest, after the warm-up, after the anaerobic priming exercise, and during recovery (Figure 1 and Figure 2). Arterialized blood samples (20 μL) were taken from a pre-warmed fingertip. The finger was always cleansed with alcohol, and the first drop of blood removed to prevent contamination of the sample. At the same time as [La] measurements, subjects were asked to evaluate their overall- and muscle fatigue using the Borg rating of perceived exertion (6 to 20-point) scale [22].

### 2.5. Measurements

Heart rate with Polar H7 sensor (Polar Electro Oy; Espoo, Finland), blood [La] concentration (EKF-Diagnostic; Barleben, Germany), body composition (Hologic Discovery DXA; Massachusetts, USA), and spirometric breath-by-breath measures (Cortex Metamax 3B, Cortex Biophysik; Leipzig, Germany) were measured and analysed. Incremental tests were performed on a Cyclus2 ergometer (RBM Elektronik-Automation GmbH; Leipzig, Germany), arm crank pre-loading exercise on a Monark Ergomedic 849E (Vansbro, Sweden) and experimental trials on a Wattbike Pro (Wattbike Limited; Nottingham, UK).

### 2.6. Statistical Analysis

Data analysis was executed using SPSS Statistics for Windows, version 23.0 (IBM Corp., Armonk, NY, USA). All data were checked for normal distribution using the Shapiro-Wilk test, visual inspection of descriptive statistics and z-score. Paired samples t-test for comparison of means and 2-way repeated measures ANOVA test for comparison of time course with *p* < 0.05 as the level of significance were applied for parametric data. Wilcoxon signed-rank test and Kruskal-Wallis test were applied in the case of nonparametric data. A prior power analysis was performed using G*Power© software (version 3.1.9.2, 2017, Heinrich-Heine-Universität Düsseldorf, Düsseldorf, Germany) for comparison between two independent means. This was based on an expected medium effect size (i.e., 0.5), an alpha criterion of 0.05, and power of 0.8. Analysis indicated that a total of 15 subjects was required to reach 0.8 statistical power with an aimed 5 s mean difference (SD of ± 5) of performance time between both interventions. Data are presented with effect size (*d*) and confidence intervals (CI). Effect size of *d* = 0.2 was considered a ‘small’, *d* = 0.5 ‘medium’ and *d* = 0.8 ‘large’ [23].

## 3. Results

### 3.1. Cycling Performance

Subjects completed the 4000 m performance task within 328.9 ± 17.4 s in MPT_high_ compared to 323.6 ± 16.2 s in MPT_low_ (*p* < 0.05; *d* = 0.7) which gave a mean difference of 5.3 ± 1.2 s (1.6%, *p* < 0.05; CI_95%_ = [0.1, 9.5]) slower time for MPT_high_. A significantly longer time was needed to recover from the strenuous 25 s high-intensity arm-crank exercise (270.6 ± 114.5 s; *p* < 0.05; *d* = 0.9) compared to the standard warm-up condition. MPT_high_ subjects started the subsequent time trial after 792.6 ± 237.2 s of recovery, compared to 522.0 ± 122.7 s in the MPT_low_ condition (CI_95%_ = [110.2, 431.0]). Data analysis showed that longer self-selected recovery time before the maximal performance tests was significantly related to a better maximal performance time in MPT_high_ (r = −0.550; *p* < 0.05). In contrast, a shorter self-selected recovery time in the MPT_low_ condition yielded a faster time trial (r = 0.619; *p* < 0.05) (Figure 3).

### 3.2. Blood Lactate Concentration

Although subjects were allowed to recover following the high-intensity arm-crank exercise without any time limit for full readiness to perform at their maximum level, the physiological variables were significantly different at the start of the time trial. The MPT_high_ priming condition significantly increased *V˙*O_2_ uptake (*p* < 0.05) and [La] concentration (*p* < 0.001), which was significantly elevated in MPT_high_ compared to MPT_low_ right before the MPT start (Table 2). Maximal [La] levels occurring in the first minute after the task completion in both conditions after the all-out test were lower in MPT_low_ compared to MPT_high_, however, these differences were not significant (Table 2). Net [La] increase was significantly lower in MPT_high_ (Table 2). Although repeated measures ANOVA did not show significantly different [La] concentrations, the [La] curve patterns for recovery were significantly different over time between both conditions (*p* < 0.001). According to [La] and mean power data, calculations showed that MPT_low_ produced more La_net_ per each W·kg^−1^, meaning more anaerobic work was performed (*p* < 0.05, *d* = 0.8).

### 3.3. V˙O_2_ and V˙CO_2_ Kinetics

A significantly higher *V˙*CO_2_ release was found already at the start in the MPT_high_ trial (0.63 ± 0.2 L·min^−1^ versus 0.54 ± 0.1 L·min^−1^; *p* < 0.05, *d* = 0.5, CI_95%_ = [−0.0, 0.2]), indicating that some buffering of acidosis from the priming was still ongoing. *V˙*CO_2_ values were lower during all time trial splits in MPT_high_ (*p* < 0.05) (Figure 4), indicating relatively decreased anaerobic energy production. A similar difference was found for *V˙*O_2_ values, since all separate 500 m splits indicated higher *V˙*O_2_ values in MPT_low_, even though *V˙*O_2_ before the start was significantly higher in the MPT_high_ condition (*p* < 0.05, *d* = 0.4) (Table 2). *V˙*O_2_ curve interactions by splits on 4000 m time trial were different and the between-group effect was significant (*p* < 0.05). According to *V˙*CO_2_ and mean power data, anaerobic power exertion was significantly lower (*p* < 0.05) during MPT_high_ at 0–500 m (*d* = 0.6), 500–1000 m (*d* = 1.0), 2000–2500 m (*d* = 0.6) and 2500–3000 m (*d* = 0.8) splits and insignificantly lower during other splits (*p* > 0.05). *V˙*CO_2_ distribution between groups was also significantly lower in MPT_high_ (*p* < 0.05). Despite the differences in *V˙*O_2_ and *V˙*CO_2_, the overall *V˙*O_2_ consumption per W·kg^−1^ did not show any significant differences between trials (*p* = 0.729). However, RER data (Figure 4) show MPT_low_ as being more anaerobic whereas MPT_high_ RER values remained lower (*p* < 0.05) during the entire race.

### 3.4. Mean Power

During the first 12 s of the 4000 m time trial mean power was 10.5 ± 2.1 W·kg^−1^ and 9.7 ± 2.5 W·kg^−1^ in MPT_low_ and MPT_high,_ respectively. There was no significant difference between both all-out start situations regarding power (*p* > 0.05, *d* = 0.5, CI_95%_ = [−1.6, 0.2]). Although total 4000 m mean power was significantly lower in MPT_high_ (*p* < 0.05, *d* = 0.7) at 5.3 ± 0.7 W·kg^−1^ compared to 5.6 ± 0.7 W·kg^−1^ in MPT_low_ (CI_95%_ = [−0.6, −0.1]) the mean power changes in both conditions were not significantly different (*p* > 0.345).

### 3.5. Ratings of Perceived Exertion

Ratings of perceived exertion (RPE) were not different at baseline (*p* > 0.751, *d* = 0.04), although after the maximal performance test subjects reported MPT_high_ as more demanding throughout recovery. During the first stage of recovery, RPE were significantly higher in MPT_high_ for muscle fatigue (from the 3rd to 7th min; *p* < 0.05, *d* = 0.5), during the second stage for overall fatigue (from the 4th to 11th min; *p* < 0.05, *d* = 0.4) and no differences during the last minutes of recovery (Figure 5).

## 4. Discussion

The main finding of this study was significantly longer (5.3 s; 1.6%) performance time when a high-intensity pre-load session was added to a traditional warm-up routine due to reduced anaerobic energy contribution indicated by a relatively smaller net [La] increase in MPT_high_.

The time trial protocol used for this study including an all-out start was adopted from Brock et al. [21] who found that an all-out pacing strategy at the beginning of the race added to a standard priming trial had the most beneficial effect on performance time. Brock et al. [21] used a 3-min submaximal 70∆ gas exchange threshold (GET) leg exercise as priming activity, and in the current study we tried to achieve the same effect using a low volume high-intensity arm-crank priming exercise.

In our study the priming effect of a 25 s high-intensity arm-crank exercise did not have beneficial effects either on performance or mean power. Subjects’ physiological response on performance trials was altered by prior priming exercise, as their respective *V˙*O_2_ and *V˙*CO_2_ curves during performance were significantly different. Moreover, during all 500 m splits *V˙*O_2_ and *V˙*CO_2_ both remained lower in the MPT_high_ condition throughout the execution of the protocol, although *V˙*O_2_ was higher just before the start. The results indicate that in MPT_high_, less absolute O_2_ was consumed, less carbon dioxide exhaled, and a larger part of work was performed aerobically. However, the total amount was insufficient to produce beneficial effects. Therefore, efficient priming mechanisms for such short-duration maximal effort are questionable, which is in line with earlier results for all-out rowing exercise [16].

Interestingly, the subjects later described the MPT_high_ performance as a “bad leg day”, uncomfortable laboratory temperature or sleepy feeling, which did not occur under the non-primed condition. This “bad leg feeling” may be attributed to the obviously decreased anaerobic energy contribution, as shown by other studies [9,18]. As all of them followed the same preparation and regimen before both performance trials, and started from a fully recovered state, we assume the arm-crank loading preceding the time trial to be the main cause. Another cause may be that subjects might had overestimated their readiness or have become additionally sensitive to external factors after having experienced the supramaximal 25 s effort. In a similar study with rowers, Purge et al. [16] also suggested that athletes who were forced to begin the time trial after 9-min recovery could not achieve their readiness to perform maximally in this study. Despite this limitation and a slower start, rowers were able to perform the second part of the distance at the same speed, thus questioning the pacing strategy [16]. Interestingly, our MPT_high_ trial cyclists’ split times never achieved the same speed compared to MPT_low_ respective splits. However, subjects were given the free will to determine their readiness level; we expected to have them fully prepared and recovered from the previous high-intensity short-duration arm-crank exercise. Accordingly, it appears that self-estimated recovery duration cannot guarantee optimal performance.

Although we considered the suggestions of previous researchers, the main findings remain unchanged. Purge et al. [16] suggested that a high-intensity priming bout could change the perception of subjects and raise a level of cautiousness that did not allow them to start at the same speed. With the intention of avoiding any self-pacing strategies, we applied the Brock et al. [21] protocol to enhance performance by all-out 12 s priming of the main 4000 m exercise bout. Brock et al. [21] presented better results for such a protocol compared to any other condition (all-out unprimed, self-paced-unprimed and self-paced-primed). In our study, subjects were asked to imitate an individual pursuit race all-out start lasting approximately 12 s. Hence distances covered during such an all-out start were 198.2 ± 14.2 m (MPT_low_) vs. 191.7 ± 16.7 m (MPT_high_) (*p* = 0.133) and mean power for the 12 s initial workout was not significantly lower in MPT_high_. The same initial power could therefore be achieved using increased aerobic mechanisms, which was surprisingly fast at fulfilling energy demands [24]. On the other hand, this power could not be sustained for the overall 4000 m time trial due to all-out pace. Purge et al. [16] mentioned that, for such a priming protocol, possibly different pacing strategies need to be developed. The slightly higher absolute systemic [La] levels may be another cause, although muscle [La] values are suggested to be lower due to inhibited glycolysis [9,18].

While O_2_ availability is crucial in maintaining the work level, prior priming has been proven to have a rapidly achievable [3] and relatively long-lasting and up to 45-min effect [25]. In earlier studies a 10-min recovery duration after priming was robustly estimated to have a beneficial effect on *V˙*O_2_ response and a substantial increase in performance [15,26]. Among our subjects, 3 out of 15 decided to start the time trial before the 10-min recovery period had elapsed, with the shortest time being 278 s. Our findings indicate that longer recovery time after the warm-up was highly correlated with better performance in MPT_high_ condition, opposite was found for MPT_low_. Meaning that in case of longer breaks during cycling sprint competition, priming even with non-sport-specific muscles could be an effective method for keeping up physiological readiness.

Previous studies have suggested that arterialized blood [La] level <5 mmol·L^−1^ could have a beneficial effect on subsequent performance as values substantially >5 mmol·L^−1^ could lead to a reduction in performance [8,15]. The results of our study show that the primed trial started at a 6.9 ± 2.1 mmol·l^−1^ [La] concentration, which was connected to lower performance. Although [La] could be used as an additional fuel for muscle contraction [27], it is also a limiting factor as decreased pH inhibits anaerobic glycolysis for subsequent performance [28]. Further studies should consider lower-intensity performance with prolonged distance and higher economy rate in order to allow body [La] clearance during some stages of the performance bout and make the priming effect more dominant.

## 5. Conclusions

In conclusion, a prior conditioning procedure of 25 s high-intensity pre-load reduced 4000 m cycling performance. Longer distances, lower intensity with less anaerobic work and a longer recovery phase should be considered in further studies to accentuate unexplained effects of prior metabolic conditioning.

## Figures and Tables

**Figure 1 sports-10-00032-f001:**
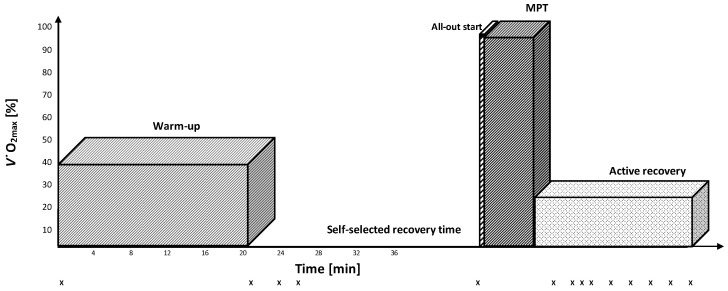
Maximal performance test with common warm-up (MPT_low_). X indicates the time of [La] measurements.

**Figure 2 sports-10-00032-f002:**
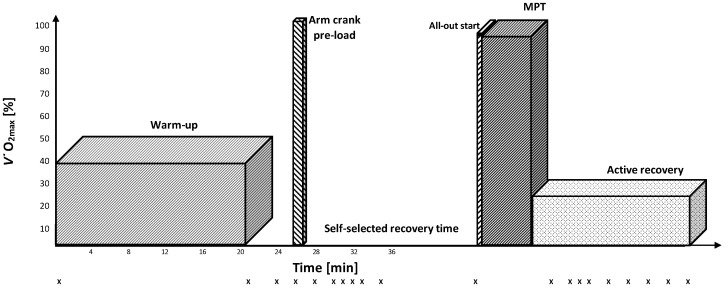
Maximal performance test protocol was added by a 25 s high-intensity all-out arm crank exercise (MPT_high_). X indicates the time of [La] measurements.

**Figure 3 sports-10-00032-f003:**
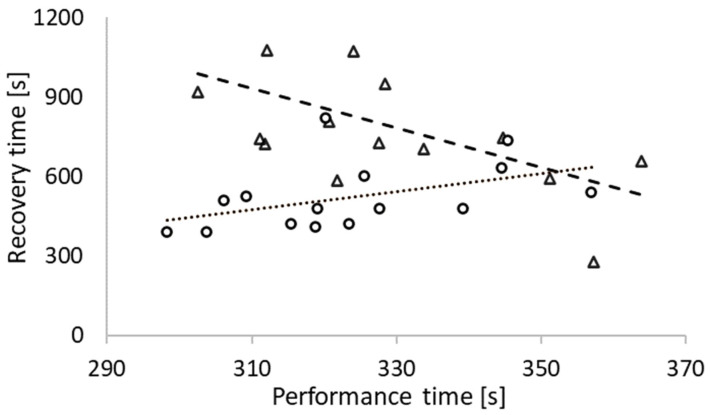
Correlation between recovery time (s) and performance time (s). Dashed line together with triangle-shaped data points indicate MPT_high_ trial (r = −0.550; *p* < 0.05) and dotted line together with circle-shaped data points MPT_low_ trial (r = 0.619; *p* < 0.05).

**Figure 4 sports-10-00032-f004:**
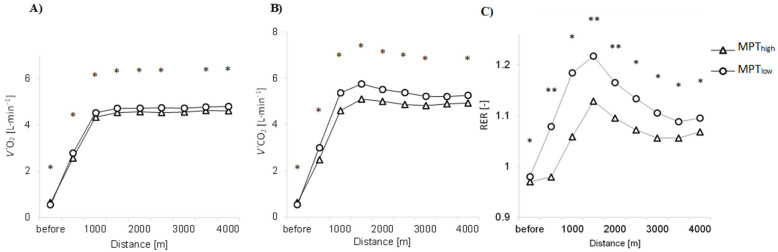
*V˙*O_2_, *V˙*CO_2_ and RER curves during MPT_low_ and MPT_high_. (**A**) *V˙*O_2_ values (L·min^−1^); (**B**) *V˙*CO_2_ values (L·min^−1^); (**C**) RER values; * *p* < 0.05; ** *p* < 0.001.

**Figure 5 sports-10-00032-f005:**
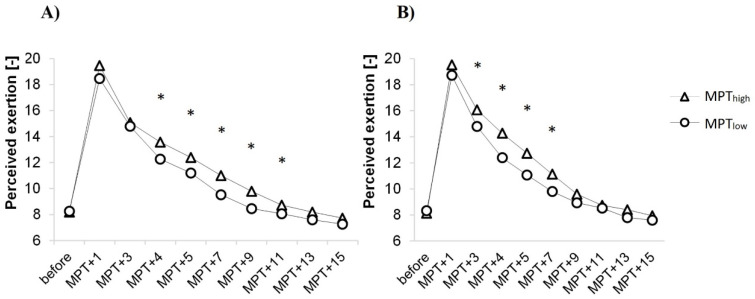
RPE values of perceived exertion. (**A**) RPE values for overall fatigue; (**B**) RPE values for muscle fatigue; MPT–maximal performance test; ‘+1’–minutes after maximal performance test; * *p* < 0.05.

**Table 1 sports-10-00032-t001:** Subjects’ characteristics (*n* = 15).

Characteristic	Mean ± SD
Age (years)	23.3 ± 3.6
Height (m)	1.8 ± 0.1
Body Mass (kg)	76.2 ± 10.0
BMI (kg·m^−2^)	23.2 ± 2.0
Fat (%)	16.1 ± 3.2
Body lean mass (kg)	58.6 ± 6.8
Leg lean mass (kg)	20.5 ± 2.1
*V˙*O_2max_ (mL·kg^−1^·min^−1^)	65.4 ± 6.7
*V˙*O_2max_ (L·min^−1^)	4.9 ± 0.5
Ventilatory threshold VT_1_ (W)	218.7 ± 33.9
Ventilatory threshold VT_2_ (W)	310.8 ± 39.3

BMI—Body mass index; *V˙*O_2max_—maximal oxygen consumption.

**Table 2 sports-10-00032-t002:** Effect of prior high-intensity warm-up on MPT (*n* = 15).

Parameter	MPT_low_	MPT_high_	*p*-Value	Effect Size (95% CI)
Initial 12 s				
Distance (m)	198 ± 14	192 ± 17	0.133	0.4 (−15.3 to 2.2)
Average power (W·kg^−1^)	10.5 ± 2.1	9.7 ± 2.5	0.104	0.5 (−1.6 to 0.2)
Peak power (W·kg^−1^)	1014 ± 219	974 ± 221	0.391	0.2 (−139.3 to 58.0)
Performance				
Time (s)	323.6 ± 16.2	328.9 ± 17.4	0.019 *	0.7 (1.0 to 9.5)
Average power (W·kg^−1^)	5.6 ± 0.7	5.3 ± 0.7	0.018 *	0.7 (−0.6 to −0.1)
La_before_ (mmol·L^−1^)	1.4 ± 0.5	6.9 ± 2.1	<0.001 **	2.5 (4.3 to 6.7)
La_net_ (mmol·L^−1^)	14.9 ± 2.2	11.1 ± 2.7	<0.001 **	1.2 (−5.5 to −2.1)
La_max_ (mmol·L^−1^)	16.3 ± 2.2	17.9 ± 2.4	0.017 *	0.7 (0.3 to 2.9)
*V˙*O_2before_ (L·min^−1^)	0.5 (0.47–0.66)	0.64 (0.51–0.78)	0.023 *	0.4 (0.03 to 0.16)
*V˙*O_2max_ (L·min^−1^)	4.9 (4.8–5.2)	4.6 (4.3–5.2)	0.013 *	0.4 (−0.4 to −0.1)
Recovery time (s)	522 ± 123	792 ± 237	0.003 *	0.9 (110.2 to 431.0)

MPT_low_—maximal performance test without prior loading; MPT_high_—maximal performance test with prior loading; CI—Confidence Interval of the Difference.* *p* < 0.05; ** *p* < 0.001.

## Data Availability

Data are contained and available within this manuscript.

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
