# Peer review of "Effect of Short-Duration High-Intensity Upper-Body Pre-Load Component on Performance among High-Level Cyclists"

_sports, 2022, doi:10.3390/sports10030032_

Round 1

Reviewer 1 Report

I congratulate the authors for this informative paper. Some of the conclusions are speculative. This should be give in terms of suggestions for future studies.

Minor-minors:

First sentence of discussion: '...of this study was significantly slower (5.3 sec; 1.6%) performance time...'. A 'slower time does not sound logic.

Reviewer 2 Report

Overall Impression

The purpose of this investigation is to examine if a high intensity bout of ark-crank ergometry added to a 20-minute cycle ergometry warm-up confers additional performance benefits to a 4000m bout of cycling, compared to a 20-minute cycle ergometry warm-up alone.  Overall, this manuscript is well written, clear, and concise.  I have mostly minor comments, which are listed below.

Introduction

  1. Page 1, line 31: Please add the word “and” before the word “higher.”
  2. Page 1, line 33: Please remove the word “sufficiency” from the end of this sentence.
  3. Page 1, line 34: Please add the word “and” before the word “increase.”
  4. Page 1, line 35: Please replace the word “were” with the phrase “have been.”
  5. Page 1, lines 36-37: I recommend removing the phrase “among athletes” from the end of the sentence.  The point is that the warm-up should not lead to fatigue, which should be true regardless of the population who is performing the warm-up, trained athlete or otherwise.
  6. Page 1, lines 37-39: This sentence is confusing.  I believe that the point of this sentence is supposed to be that there are multiple factors that influence how a particular conditioning activity performed during warm-up will affect subsequent performance. These factors include, but are not limited to athlete-specific factors (e.g. muscular strength and resistance training experience), the specific warm-up activities performed, and the length of the rest period that occurs between the warm-up activity and the exercise performance activity.  Therefore, optimal performance during a particular athletic bout depends on the warm-up intensity that takes these factors into consideration.
  7. Page 1, line 40: Please replace the word “already” with the word “in” at the beginning of this sentence.
  8. Page 1, lines 42-43: In this sentence, you are describing that numerous studies have been published in the decades since the study published by Klausen et al. in 1972.  However, there is no reference citation at the end of this sentence.  Please add the pertinent reference citation(s) to the end of this sentence.
  9. Page 1, line 42: Please clarify that these studies examined increased VO2 kinetics.
  10. Page 1, lines 43-45: Please clarify what you mean by a “standardized approach.”  The way that this sentence is worded, it implies that Carter et al. (reference #7) suggests a standardized approach to the pre-performance warm-up.  I don’t believe that’s what the study’s authors are implying in their conclusions. 
  11. Page 2, line 46: Please define “traditional warm-up process.”
  12. Page 2, lines 49-51: For clarity, I advise rewording this sentence to say, “In contrast, more general warm-up approaches can impact muscles globally without raising local muscle 50 acidosis or using glycogen stores [12].
  13. Page 2, line 52: Please add the word “approach” after the word “pre-conditioning.”
  14. Page 2, lines 60-61: I believe that you can replace the phrase, “As a main reason to be not effective improving performance” and replace it with the word “Furthermore”. 
  15. Page 2, lines 66-67: You begin this paragraph with the phrase, “As previously mentioned…”  However, you really have not discussed pacing strategies, except for mentioning pacing in the final sentence of the preceding paragraph. Therefore, I advise adding more text to the beginning of this current paragraph to more clearly introduce the reader to pacing and its benefits, as it relates to the rationale for your study.

Materials and Methods

  1. Page 2, lines 83-84: For clarity, I recommend that you reword the last sentence of this paragraph to say, “In addition, subjects were asked to refrain from maximal exercise for at least 3 days before each laboratory visit.”

Results

  1. Page 4, lines 166-167: I recommend placing a period after “(p<0.05)” in line 166.  Then, begin a new sentence and say, “In contrast, a shorter self-selected recovery time in the MPT-low condition yielded a faster time trial (r= 0.619; p<0.05) (Fig. 2).”  Since I’ve recommended that you take your currently-existing Figure 1 and make that into two figures, the currently-existing Figure 2 would become Figure 3.
  2. Page 6, line 192: I recommend replacing the word “inhibited” with the phrased “relatively decreased.”  Using the word “inhibited” implies that anaerobic metabolism was not occurring at all, which is not true.  Instead, anaerobic energy production was decreased during the MPT-high condition, relative to the MPT-low condition.

Discussion

  1. Page 7, line 229: Again, I recommend replacing the word “inhibited with the phrase “a relatively smaller net [La] increase in MPT-high. To say that there was an inhibited net [La] increase for the MPT-high condition implies that [La] did not increase, which is not true.  [La} did indeed increase during the MPT-high condition, just not to the same extent as it did for the MPT-low condition.
  2. Page 7, line 250: Again, I recommend replacing the word “inhibited” with the word “decreased.”

Tables and Figures

  1. Table 1: I recommend adding percent body fat to this table. 
  2. Figure 1: I recommend that you create two figures, one to depict the protocol involving the MPT-high condition and another to depict the MPT-low condition.  Collapsing both conditions into one figure may be confusing to some readers.
  3. Figure 2: Please also add to the figure legend that the triangle-shaped data points represent the MPT-low trials, and the circle-shaped data points represent the MPT-high trials. Even though you’ve stated the correlation coefficients and p-values in the Results on page 4, I recommend that you add them to the figure legend as well.
